# Primary Retroperitoneal Carcinomas: New Insights into Pathogenesis and Clinical Management in Comparison with Ovarian Carcinomas and Carcinoma of Unknown Primary

**DOI:** 10.3390/cancers15184614

**Published:** 2023-09-18

**Authors:** Isao Otsuka

**Affiliations:** Department of Obstetrics and Gynecology, Kameda Medical Center, Kamogawa 296-8602, Japan; otsuka.isao@kameda.jp

**Keywords:** retroperitoneal carcinoma, ovarian carcinoma, carcinoma of unknown primary, pathogenesis, endosalpingiosis, endometriosis, spontaneous regression, vanishing cancer

## Abstract

**Simple Summary:**

Primary retroperitoneal carcinomas are very rare tumors. Their pathogenesis remains unknown but may be associated with that of ovarian carcinomas, considering the similarity in morphology and gender preference. Mucinous carcinoma, which develops in both women and men, may originate in both primordial germ cells and Walthard cell nests that may be derived from the fallopian tube. Serous carcinomas may be associated with endosalpingiosis and a remnant Müllerian tract. Endometrioid and clear cell carcinomas appear to be associated with extraovarian endometriosis. Additionally, both carcinomas in the retroperitoneal lymph nodes may be metastatic diseases from endometrial and/or renal cell cancers that regress spontaneously (carcinoma of unknown primary). Surgery is the cornerstone of treatment, but the necessity of chemotherapy may depend on histological subtype. Further studies are necessary, in particular studies on endosalpingiosis, which is a poorly understood condition, although associated with the development of both serous and mucinous carcinomas.

**Abstract:**

Primary retroperitoneal carcinomas are very rare tumors. Their pathogenesis remains unknown but may be associated with that of ovarian carcinomas, considering the similarity in morphology and gender preference. Although metaplasia of coelomic epithelium is the most widely accepted theory, the pathogenesis of retroperitoneal carcinomas may differ by histologic subtype, like ovarian carcinomas. Mucinous carcinoma, which develops in both women and men, may originate in both primordial germ cells and Walthard cell nests that may be derived from the fallopian tube. Serous carcinomas may be associated with endosalpingiosis, the presence of fallopian tube-like epithelium outside the fallopian tube, and a remnant Müllerian tract. Endometrioid and clear cell carcinomas appear to be associated with extraovarian endometriosis. Additionally, both carcinomas in the retroperitoneal lymph nodes may be metastatic diseases from endometrial and/or renal cell cancer that regress spontaneously (carcinoma of unknown primary). Retroperitoneal carcinomas are difficult to diagnose, as they have no characteristic symptoms and signs. Surgery is the cornerstone of treatment, but the necessity of chemotherapy may depend on histological subtype. Further studies are necessary, in particular studies on endosalpingiosis, as endosalpingiosis is a poorly understood condition, although it is associated with the development of both serous and mucinous carcinomas.

## 1. Introduction

In the retroperitoneal space, many types of tumors develop, both primary and metastatic [1]. Primary retroperitoneal malignant neoplasms account for only 0.1–0.2% of all malignancies [2], and most of these neoplasms are mesenchymal, neurogenic or lymphatic, i.e., those originating from tissues in the retroperitoneal space [3]. In addition, primary epithelial neoplasms can also develop in this space. Of these, primary retroperitoneal epithelial carcinomas are very rare.

The pathogenesis of primary retroperitoneal carcinomas remains unknown. Their pathogenesis, however, may be associated with that of ovarian carcinomas, as histological subtypes of retroperitoneal carcinomas are the same as those of ovarian carcinomas. In addition, primary retroperitoneal carcinomas almost exclusively develop in women, excluding mucinous tumors that develop in both women and men. Recently, there has been a paradigm shift in our understanding of the pathogenesis of ovarian carcinomas and their origins [4,5]. Ovarian carcinomas are a group of neoplasms that are essentially distinct diseases [6]. Additionally, most ovarian carcinomas may be imported diseases, i.e., they stem from Müllerian-derived extraovarian cells that involve the ovary secondarily [7]. Similarly, retroperitoneal carcinomas may be imported diseases arising from cells of extra-retroperitoneal origin. Moreover, some types of carcinomas of unknown primary may be classified as primary retroperitoneal carcinoma, considering that carcinomas of unknown primary are often observed only in the retroperitoneal space. This narrative review explores the pathogenesis of primary retroperitoneal carcinomas based on the knowledge of ovarian carcinogenesis. In addition, the clinical management of retroperitoneal carcinomas is discussed.

## 2. Classification of Retroperitoneal Carcinomas

Retroperitoneal carcinomas can be classified according to histological characteristics and the site of the disease.

### 2.1. Histological Subtype

Histological subtypes of retroperitoneal epithelial carcinomas include mucinous [8,9,10,11,12,13], serous [14,15,16,17,18,19,20,21,22,23,24,25,26,27,28,29], endometrioid [30,31,32,33,34], clear cell [35,36,37,38], and carcinosarcoma [39,40,41]. Morphologically, mucinous, serous, and endometrioid tumors resemble the phenotype of the colon, fallopian tube, and endometrium, respectively [42]. Clear cell carcinoma develops in the female genital tract (the ovary, endometrium, cervix, and vagina) as well as in the kidney, and may have similar morphological features regardless of the site of origin [43].

Mucinous carcinoma is the most common subtype of primary retroperitoneal carcinomas. More than 70 cases have been reported in the literature (Table 1) [8,9]. Of retroperitoneal mucinous tumors, carcinomas and borderline tumors are more common than benign cystadenoma [10]. Although retroperitoneal mucinous tumors develop in both women and men, a previous study reported that 94% of mucinous carcinomas develop in women (Table 1) [8]. Median age at diagnosis in women was significantly younger than that in men (42.0 years vs. 62.2 years) [8]. Primary retroperitoneal mucinous carcinomas diagnosed during pregnancy have been reported [11,12,13]. Retroperitoneal serous carcinoma is a very rare occurrence, with 16 cases having been reported (Table 2). All but one case developed in women. The ages range from 11 to 75 years (median 58 years), with only one patient developing primary retroperitoneal serous carcinoma prepubertally. Histologically, both low-grade and high-grade serous carcinomas were observed. Of note, a few cases reported as primary retroperitoneal serous carcinoma might not be primary but lymph node metastasis from adnexal or peritoneal carcinoma [26], although there is a possibility that they may be double synchronous primaries. Retroperitoneal endometrioid and clear cell carcinomas are even rarer, as only five and four cases have been reported, respectively (Table 3) [30,31,32,33,34,35,36,37,38]. Whereas endometrioid carcinomas developed only in women, two of the four clear cell carcinomas developed in men [36,38]. These retroperitoneal carcinomas may be diagnosed as carcinomas of unknown primary [38]. Mixed epithelial carcinomas (Table 3) [37,44] and carcinosarcoma [39,40,41] also develop in the retroperitoneal space.

### 2.2. Site of Disease

Retroperitoneal carcinomas develop in the lymph nodes and/or extranodal sites. In cases of huge tumors, this classification may not be possible. When carcinomas are found in the lymph nodes alone, they may be diagnosed as carcinoma of unknown primary [33].

## 3. Previously Postulated Pathogenesis of Retroperitoneal Carcinomas

Although the pathogenesis of primary retroperitoneal carcinomas has not been elucidated, several hypotheses have been postulated [45,46,47,48,49]. The most widely accepted theory is metaplasia of coelomic epithelium [14,15,16,46,47,48]. Other possible origins include extra-ovarian endometriosis, teratoma or primordial germ cell, ectopic ovarian tissue, and intestinal duplication (enterogenic cyst) [17,21].

The coelomic epithelium gives rise to the peritoneum [50], and invagination of the peritoneum results in inclusion cysts in a retroperitoneal space [45]. The Müllerian duct, which eventually forms the fallopian tube, endometrium, endocervix, and the upper third of the vagina, is known to form by invagination of the coelomic epithelium [51]. The coelomic epithelium covering cysts may subsequently undergo metaplasia [52]. Malignant transformation of endometriosis, which is a gynecological disease defined by the histological presence of endometrial glands and stroma outside the uterine cavity [53], is a rare but well-known complication. Migratory arrest of primordial germ cells is thought to be the basis of a teratoma in the retroperitoneal space. Monodermal variants of teratomas may be an origin of retroperitoneal carcinomas [47,48]. Ectopic ovarian tissue and intestinal duplication may be the origins of retroperitoneal carcinomas [47,48]. However, the intestinal duplication hypothesis seems unlikely, as enterogenic cysts are gut-like and they consist of one or two layers of smooth muscle and gut mucosa [54].

## 4. Pathogenesis of Ovarian Carcinomas

As stated above, ovarian carcinomas are a group of distinct diseases, exhibiting a wide range of morphological features, clinical manifestations, and tumor behaviors [55]. In addition, they are different in terms of epidemiological and genetic risk factors, precursor lesions, and molecular events during oncogenesis [6].

Traditionally, all of the four major subtypes of ovarian carcinomas, i.e., serous, endometrioid, clear cell, and mucinous tumors, were thought to derive from the common origin, i.e., ovarian surface epithelium [56,57]. The ovarian surface epithelium, which is derived from the coelomic epithelium, is the pelvic peritoneum that overlies the ovary and lines ovarian epithelial inclusion cysts [58,59]. In contrast to this traditional view, most ovarian carcinomas are now believed to derive from endometrial tissue, fallopian tube tissue, and germ cells [60]. Thus, most ovarian carcinomas are primarily imported from either endometrial or fallopian tube epithelium, unlike other human cancers in which all primary tumors arise de novo [61].

### 4.1. Mucinous Carcinoma

Among the four major subtypes of ovarian carcinomas, mucinous carcinoma is distinct from other histologic subtypes [62]. Different from non-mucinous carcinomas that morphologically resemble epithelia originating in the Müllerian duct, mucinous carcinoma resembles the colon epithelium. Risk/protective factors for ovarian carcinomas include parity, breast-feeding, oral contraceptive use, and family history of ovarian/breast cancer [63]. However, these factors were not associated with the development of mucinous carcinomas [64,65,66].

Mucinous ovarian carcinoma develops via a sequence from benign tumor through borderline tumor to invasive cancer [67,68]. In contrast to non-mucinous ovarian carcinomas that are associated with ovulation and menstruation [69,70,71], a mucinous ovarian carcinoma develops in premenarchal girls [72], suggesting that some mucinous tumors have common features with germ cell tumors. Smoking [64,65,66] and heavy alcohol consumption [73] that is associated with colorectal cancers [74] are risk factors for mucinous carcinomas. 

Recently, mucinous carcinomas are suggested to originate from either teratomas or Brenner tumors [75]. Mucinous cystic tumors sometimes arise in the context of mature cystic teratomas and other primordial germ cell tumors [76,77,78]. Patients with teratoma-associated mucinous tumors were significantly younger than patients with Brenner tumor-associated mucinous tumors (43 vs. 61 years) [75]. Interestingly, teratomas and Brenner tumors give rise to different subtypes of mucinous ovarian tumors [75]. The presence of mucinous tumors within Brenner tumors has been reported to be up to 33% [75], and molecular genetic analysis indicated that mucinous and Brenner tumors have a shared origin [79]. Endosalpingiosis, i.e., the presence of fallopian tube-like epithelium outside the fallopian tube, is also associated with mucinous tumors [80,81]. A recent study using genetic analyses indicated that mucinous ovarian tumors of germ cell origin may be rare [82].

### 4.2. Serous Carcinoma

Most serous carcinomas develop from the fallopian tube epithelium [4,5]. Serous carcinomas, the most common histological subtype of epithelial ovarian carcinoma, are divided into high-grade and low-grade carcinomas, as they are different in terms of pathogenesis, disease spread pattern, and prognosis [5]. A significant proportion of high-grade serous carcinoma develops in the secretory epithelial cells of the tubal fimbria [83,84]. Risk-reducing salpingo-oophorectomy has been performed in women with germline *BRCA* mutation who are at risk for developing high-grade serous carcinoma, since high-grade serous carcinoma cannot be detected early by screening using transvaginal ultrasound [85,86]. Pathological evaluation of the removed ovaries and fallopian tubes revealed that a precursor lesion of high-grade serous carcinoma, serous tubal intraepithelial carcinoma, is located in the fallopian tube epithelium [87]. This discovery led to a paradigm shift in our understanding of the origin of ovarian carcinomas. However, 12–40% of high-grade serous carcinomas are reported to derive from ovarian surface epithelium [4,88,89]. Low-grade serous carcinoma, which develops from benign ovarian cyst (cystadenoma) through borderline tumor to invasive carcinoma, is also of tubal origin [90]. Ovarian epithelial inclusions, from which serous cystadenoma develops, are likely to derive from the fallopian tube through an intraovarian endosalpingiosis rather than through Müllerian metaplasia from ovarian surface epithelium [90]. However, recent studies reported that intraovarian endosalpingiosis may not arise as a consequence of detachment and implantation of tubal epithelium [91,92].

### 4.3. Endometrioid Carcinoma

Endometrioid carcinoma, as well as clear cell carcinoma, is derived from endometriosis. Endometriosis, in particular endometriotic cyst, is a precursor to endometriosis-associated ovarian carcinomas [93,94,95], as the molecular genetic alterations present in endometriosis-associated carcinomas can be found in adjacent endometriosis lesions [95]. Ovarian endometriosis occurs primarily as a result of retrograde menstruation and implantation of endometrial tissue fragments in ovarian inclusion cyst [70,96]. A recent study suggested that endometrioid carcinomas derive from cells of the endometrial secretory cell lineage, whereas clear cell carcinomas derive from cells of the endometrial ciliated cell lineage [97]. Of note, endocervical-like (Müllerian) mucinous tumors are now classified as a subtype of endometrioid tumors [98].

### 4.4. Clear Cell Carcinoma

Although clear cell carcinoma is another endometriosis-associated carcinoma in women, it develops not only in ovarian endometriotic cysts but also the kidney [99,100,101]. Clear cell carcinoma of the ovary shares many histologic features with renal cell carcinoma, especially translocation-associated renal cell carcinoma, hence the determination of the origin may not be possible based on the morphology alone [101]. Using immunohistochemical markers, ovarian clear cell carcinoma can be distinguished from renal clear cell and translocation-associated carcinoma [101]. Interestingly, however, clear cell carcinomas showed remarkably similar gene expression patterns regardless of their origin [102].

### 4.5. Carcinosarcoma

Carcinosarcomas are biphasic neoplasms that are composed of an admixture of malignant epithelial and mesenchymal elements. Most of the gynecological (uterine and ovarian) carcinosarcomas have a monoclonal origin [103], as the mutations identified were present in both carcinomatous and sarcomatous components [104]. Carcinomatous cells transform into sarcomatous cells [105,106,107]. Heterogeneous molecular features observed in uterine carcinosarcomas resemble those observed in endometrial carcinomas, with some showing endometrioid carcinoma-like and others showing serous carcinoma-like mutation profiles [104]. Frequent mutations observed in uterine carcinosarcomas are similar to endometrioid and serous uterine carcinomas [108]. Thus, carcinosarcoma may derive from endometrioid or serous carcinoma.

## 5. Newly Proposed Hypothesis on Pathogenesis of Retroperitoneal Carcinomas

Like the pathogenesis of ovarian carcinomas, the pathogenesis of retroperitoneal carcinomas may differ by histologic subtype, considering their similarity in morphology and gender preference. A major shortcoming of the previously proposed hypotheses is that they consider all histologic subtypes to have a common pathogenesis. Although the theory of coelomic metaplasia cannot be excluded, new theories of the pathogenesis of retroperitoneal carcinomas reflecting the new paradigm of ovarian carcinogenesis can be postulated (Figure 1).

### 5.1. Mucinous Carcinoma

Origins of retroperitoneal mucinous tumors may be primordial germ cells and Walthard cell nests like ovarian mucinous tumors. Primary retroperitoneal mucinous carcinomas have immunohistochemical staining patterns similar to those of ovarian mucinous tumors [109], and mucinous tumors develop in both women and men, as well as teratoma that is of germ cell origin. The migration of primordial germ cells in the embryo starts from the dorsal wall of the yolk sac. Then primordial germ cells migrate into the midgut and hindgut, passing through the dorsal mesentery into the gonadal ridges. Thus, mucinous tumors can arise from primordial germ cells that stopped anywhere in the retroperitoneal space. Mucinous carcinomas arising from a retroperitoneal mature cystic teratoma have been reported [110,111].

Mucinous tumors also arise from sites of transitional cell metaplasia, known as Walthard cell nests [112,113], from which Brenner tumors are believed to derive. The tuboperitoneal junction cells that undergo transitional cell metaplasia and invaginate into the paratubal or ovarian surface form Walthard cell nests [112,114]. Walthard cell nests may derive from the fallopian tube, since fallopian tube secretory cells, transitional metaplasia, Walthard cell nests, and the epithelial component of Brenner tumors share a similar immunohistochemical profile [114]. Thus, mucinous tumors may also arise from the embryological remnant of the Müllerian duct [115]. Some mucinous tumors may develop from a Brenner tumor, as a shared clonal relationship between the mucinous and Brenner tumor components has been demonstrated [116]. In these cases, the Brenner tumor component becomes compressed and obliterated by an expanding mucinous neoplasm [116].

### 5.2. Serous Carcinoma

The pathogenesis of retroperitoneal serous carcinomas appears to be associated with endosalpingiosis. Although endosalpingiosis and endometriosis occur concurrently in 34% of endosalpingiosis cases [117], endosalpingiosis is not a variant of endometriosis as they have different clinical presentations [117]. Endosalpingiosis is not significantly associated with infertility or chronic pain and its incidence increases with age [118], with 40% of endosalpingiosis cases being postmenopausal [117].

Endosalpingiosis, as well as retroperitoneal serous carcinomas, can be divided into non-nodal and nodal lesions. Non-nodal endosalpingiosis is observed in the peritoneum and subperitoneal tissues [117]. In the peritoneal cavity, endosalpingiosis is observed in the ovary, fallopian tube, uterus, omentum, small bowel, and sigmoid colon [117]. Its prevalence is 7.6% in women undergoing laparoscopic surgery for gynecologic conditions [119]. However, a recent study reported that, among benign-appearing adnexal lesions evaluated using the Sectioning and Extensive Examining-Fimbria protocol, the prevalence of endosalpingiosis was 22% [118]. Müllerian inclusions in lymph nodes, which are usually identical to endosalpingiosis [120], occur in pelvic or para-aortic lymph nodes in 11–23% of women [120,121,122].

The pathogenic mechanism of endosalpingiosis remains unclear. Two major theories are postulated, namely, the tubal escape theory and the Müllerian metaplasia theory [92,123]. In the tubal escape theory, shed tubal epithelium either implants on the peritoneal surface or disseminates via lymphatics into a lymph node [123]. Tubal epithelia are easily sloughed off by tubal inflammation, tubal lavage, or salpingectomy [122]. In the Müllerian metaplasia theory, latent cells present in ectopic locations outside the Müllerian tract (fallopian tube, endometrium, endocervix) retain the capacity for forming benign tubal-type glands [92]. Endosalpingiosis that arises from the embryological remnant of the Müllerian duct may be an origin of retroperitoneal serous carcinoma, especially in men [123]. In addition, endosalpingiosis may be caused by the dislocation of primitive tubal tissue outside the fallopian tube during organogenesis, as well as endometriosis [53].

Endosalpingiosis appears to be a latent precursor to low-grade pelvic serous carcinomas [124] and prone to neoplastic transformation, as it represents cancer driver mutations that are observed in ovarian low-grade serous tumor [125]. Endosalpingiosis is associated with ovarian and uterine cancers [80]. Patients with endosalpingiosis and ovarian cancer have the increased prevalence of serous borderline, invasive mucinous, and clear cell carcinomas [80]. Atypical endosalpingiosis, i.e., lesions exhibiting architectural changes and/or cytologic atypia intermediate between endosalpingiosis and serous borderline tumor, is observed either mesothelial or submesothelial in location [126]. From lymph node inclusions, low-grade serous tumors (low-grade serous carcinoma and borderline serous tumor) have been reported to develop [120,122,127,128].

Although low-grade serous tumor and high-grade serous carcinoma are two distinct diseases, low-grade serous carcinoma may transform into high-grade serous carcinoma in rare cases [129,130,131]. A morphologic continuum between the low-grade tumors and high-grade carcinoma was observed in four cases [129]. The same mutation of *KRAS* was found in both the serous borderline tumor and the high-grade serous carcinoma components of the tumor in two cases [129]. In addition, *TP53* and *NRAS* mutations are important in high-grade transformation [131,132]. Of note, endosalpingiosis is associated with germline *BRCA* mutation [124].

Serous carcinoma and endosalpingiosis can develop in men [25,123]. These lesions may originate from Müllerian cell rests remaining after regression of the Müllerian duct. During fetal development, before the secretion of anti-Müllerian hormone by the fetal testes, the Müllerian ducts develop in male fetuses [133]. Thus, Müllerian tissue may remain in a male body. Endometriosis can also develop in men who had been treated with prolonged estrogen therapy for prostate cancer or diagnosed as cirrhosis [134,135]. In addition, a man with persistent Müllerian duct syndrome, a rare form of internal male pseudo-hermaphroditism characterized by the presence of a rudimentary Müllerian duct in an otherwise phenotypically and genotypically normal man [136], was reported to develop clear cell carcinoma of the remnant uterus [137].

A few cases of retroperitoneal serous carcinoma may not be primary but metastasis from adnexal or peritoneal carcinoma [18,20,26]. Instead, these patients may be cancer-prone individuals and they developed double primary carcinoma as they had a *BRCA* mutation-related cancer, breast and peritoneal. In ovarian serous borderline tumors, nodal foci of serous borderline tumor have been suggested to derive from nodal endosalpingiosis independent of ovarian tumors [122].

### 5.3. Endometrioid Carcinoma

Retroperitoneal endometrioid carcinoma appears to arise from endometriosis of the urinary tract and lymph nodes. Endometrioid tumors are the majority of extragonadal endometriosis-associated carcinomas [138]. Tumors arising in endometriosis are predominantly low-grade and confined to the site of origin [138]. Pelvic surgery may affect the development of retroperitoneal endometrioid carcinoma. In addition, estrogen replacement therapy may promote malignant transformation in endometriosis [139,140]. Two patients who underwent hysterectomy and bilateral salpingo-oophorectomy developed retroperitoneal endometrioid carcinoma in the peri-ureter site after estrogen replacement therapy (Table 3) [30,31].

Endometrioid carcinoma may also arise in the lymph nodes and it appears to develop from nodal endometriosis. Endometriosis is believed to disseminate via the lymphatic system [141]. Metastatic endometriosis lesions in pelvic sentinel lymph nodes were present in women with ovarian and/or pelvic endometriosis [142]. Interestingly, both of the patients who developed endometrioid carcinoma in the pelvic lymph nodes had Lynch syndrome [33,34]. Lynch syndrome is an autosomal dominant cancer predisposition syndrome caused by germline mutations in DNA mismatch repair (MMR) genes, i.e., *MLH1*, *MSH2*, *MSH6*, and *PMS2*, related to an increased risk of developing colorectal, endometrial, and ovarian cancers [143]. Thus, the malignant transformation of endometriosis in the lymph nodes in women with Lynch syndrome may be likely to occur.

Instead, endometrioid carcinoma in the retroperitoneal lymph nodes may be metastatic lesions from endometrial carcinoma that regresses spontaneously. Pelvic lymph node metastasis was observed in 3.5% of endometrioid endometrial carcinomas with superficial myometrial invasion, even in carcinoma without myometrial invasion [144]. In endometrial cancer, especially endometrioid carcinoma, the vanishing cancer phenomenon has been known to occur [145,146]. A cause of spontaneous regression may be an anti-tumor immune response. Among various cancers, MMR deficiency is most often observed in endometrial carcinoma [147]. MMR-deficient endometrial carcinomas, in particular Lynch syndrome-associated carcinoma, have a high number of tumor-infiltrating lymphocytes [148], as these cancer cells accumulate somatic mutations that are associated with neoantigen production and result in strong immunoreactions [149]. Recently, the spontaneous regression of MMR-deficient colon cancer in three patients was reported [150,151,152]. In one of these patients, a metastatic lymph node was histologically confirmed while a cancerous lesion in the colon, which was observed in a biopsy specimen, could not be detected in the colectomy specimen [151].

### 5.4. Clear Cell Carcinoma

Clear cell carcinoma is another endometriosis-associated carcinoma and may arise in both retroperitoneal endometriosis and lymph nodes. A case who underwent supracervical hysterectomy and bilateral salpingo-oophorectomy and received 4 years of estrogen therapy developed clear cell carcinoma arising in retroperitoneal endometriosis [35]. This case may also suggest that estrogen replacement may be a risk factor for developing endometriosis-associated cancer [139]. In men, retroperitoneal clear cell carcinoma may be a metastatic disease from renal clear cell carcinoma in the absence of a primary tumor in the kidneys (carcinoma of unknown primary) (Table 3) [36,38]. In these cases, primary renal cell carcinoma might regress spontaneously [153,154].

### 5.5. Carcinosarcoma

Retroperitoneal carcinosarcoma, as well as uterine carcinosarcoma [108], may derive from serous or endometrioid carcinoma. In ovarian carcinosarcoma, frequently encountered epithelial components are serous, endometrioid, or undifferentiated carcinomas [155]. Of the three retroperitoneal carcinosarcomas, one was associated with endometriosis that is known to be associated with the development of endometrioid carcinoma [41]. Another case developed retroperitoneal carcinosarcoma seven years after surgery for bilateral serous ovarian carcinoma. In that case, retroperitoneal carcinosarcoma may derive from an occult metastatic lesion of previous serous ovarian carcinoma. The remaining case might arise from germ cells in the remnants of the urogenital ridge [39]. Carcinosarcoma can also arise in a cystic teratoma [156].

## 6. Retroperitoneal Carcinoma as a Part of Carcinoma of Unknown Primary

Part of carcinomas of unknown primary may be a primary retroperitoneal carcinoma. Carcinoma of unknown primary represents a heterogeneous group of metastatic tumors for which a standard diagnostic work-up fails to identify the site of origin at the time of diagnosis [157]. Patients with carcinoma of unknown primary are categorized into two prognostic subgroups according to their clinicopathologic characteristics. The majority of patients with carcinoma of unknown primary (80–85%) belong to unfavorable subsets. In this subset, two prognostic groups are identified according to LDH level and the performance status [158,159]. The favorable risk cancer subgroup (15–20%) includes patients with neuroendocrine carcinomas of unknown primary. Very recently, new favorable subsets of carcinoma of unknown primary seem to emerge, including colorectal, lung, and renal carcinoma of unknown primary [159]. Although the incidence of carcinoma of unknown primary is decreasing, probably due to advanced diagnosis, it accounts for 2–5% of newly diagnosed advanced malignancies [158]. Metastatic adenocarcinoma is the most common histopathology [160].

Carcinoma in the lymph node diagnosed as a carcinoma of unknown primary may be a primary carcinoma that derives from endosalpingiosis or endometriosis in the node. Serous carcinomas in the inguinal node have been reported [161,162]. In addition, atypical hyperplasia with noninvasive, well-differentiated endometrioid carcinoma and high-grade serous carcinoma within a focus of endometriosis in an inguinal mass in a young woman has been reported [163]. In addition, cases with a primary tumor regressing spontaneously and lymph node metastasis remaining may be diagnosed as a carcinoma of unknown primary.

## 7. Clinical Management

### 7.1. Diagnosis

Retroperitoneal carcinomas are difficult to diagnose, as they have no characteristic symptoms and signs. They cannot be diagnosed until tumors cause symptoms or become palpable. Symptoms commonly associated with retroperitoneal mucinous tumors include abdominal pain and palpable mass (Table 1) [8,164]. For retroperitoneal serous carcinomas, the most common symptom was abdominal pain (Table 2) (6/17, 35%). Of note, four patients (4/17, 24%) were asymptomatic. Regular check-ups using imaging studies or tumor marker evaluation may incidentally detect retroperitoneal carcinomas in asymptomatic patients.

Making a correct preoperative diagnosis of retroperitoneal carcinomas is difficult and a definitive diagnosis can only be made at surgery. However, cross sectional imaging studies, such as computed tomography and magnetic resonance imaging, can demonstrate important characteristics of the tumors, such as lesion location, size, shape, and thickness of a wall. However, the precise localization of the lesions, determination of the extent of invasion, and characterization of the histological subtype are difficult [165]. 18F-fluorodeoxyglucose positron emission tomography/computed tomography (PET/CT) may be useful for the discrimination of malignant and benign masses [29,166]. In addition, PET/CT is an effective method for the detection of unknown primary tumors [167]. For the diagnosis of retroperitoneal serous carcinoma, the evaluation of serum CA-125 appears to be useful, as 90% of the cases had an elevated level (Table 2).

### 7.2. Treatment

Surgery is the cornerstone of the treatment of retroperitoneal carcinoma. A surgical approach is necessary to definitively diagnose these tumors. Complete tumor resection, in particular without rupture of the capsule, appears to be required to improve outcomes [168,169]. In young women with a desire to spare fertility, excision of only the tumor should be considered, when the tumor remains confined to the retroperitoneum and is well- to moderately-differentiated [170]. In patients with non-mucinous retroperitoneal carcinoma, in particular serous carcinoma, a thorough exploration of the abdominal cavity is mandatory, as the coexistence of diaphragmatic implants and peritoneal recurrence after tumor resection has been reported [18,19]. Bilateral salpingo-oophorectomy with hysterectomy may be performed in patients with non-mucinous carcinomas, as concurrent adnexal serous carcinoma including intraepithelial carcinoma may exist [20,26]. Laparoscopic resection of retroperitoneal tumors is feasible, but the long-term safety is not fully evaluated [24,28,163]. For women who developed retroperitoneal tumors during pregnancy, the timing of surgery should be determined by both malignancy risk and obstetric considerations [168].

The necessity for chemotherapy after surgery appears to depend on histological subtype. Mucinous carcinoma is known to be relatively chemoresistant [171], thus adjuvant chemotherapy may not be necessary when the tumor is resected completely. In non-mucinous carcinomas, a retroperitoneal tumor may not be primary but metastasis from clinically undetectable primary tumors, thus adjuvant treatments should be considered. As the majority of retroperitoneal serous carcinomas are high-grade, adjuvant chemotherapy consisting of paclitaxel and carboplatin should be administered [20,22,23,25,26,29], as well as its ovarian counterpart. The role of adjuvant radiotherapy in the treatment of retroperitoneal carcinomas is uncertain.

### 7.3. Prognosis

Although long-term follow-up data are not available for many cases of retroperitoneal carcinomas, treatment outcomes appear to differ by histological subtype. Mucinous tumors are usually associated with a favorable prognosis, except those with mural nodules that are often diagnosed as sarcoma, anaplastic carcinoma, or carcinosarcoma [172,173]. Five-year disease-specific survival of patients with retroperitoneal mucinous carcinoma was 75% [8]. Adjuvant chemotherapy was associated with a reduced survival in mucinous carcinoma [8,9], suggesting that patients at high-risk for recurrence were more likely to receive chemotherapy. In contrast, patients with retroperitoneal serous carcinoma have poor survival. Two- and 5-year disease-free survival was 53% and 18%, respectively (Figure 2). However, retroperitoneal serous carcinoma of nodal type may have a favorable survival [29], similar to patients with serous carcinoma of the ovary, fallopian tube, or peritoneum, who have lymph node metastasis and minimal peritoneal disease [174,175]. Extragonadal endometriosis-associated carcinomas are associated with a favorable survival when the tumor was confined to the extragonadal site of origin, with a 5-year survival of 100% [138]. For retroperitoneal carcinomas, of the eight patients with endometrioid or clear cell tumors, only one patient who may have a primary renal cell carcinoma developed recurrence (Table 3).

## 8. Conclusions

Although many hypotheses are discussed in this review, the exact pathogenesis of retroperitoneal carcinomas remains to be elucidated. Retroperitoneal carcinomas may not originate in the coelomic epithelium but in the Müllerian epithelium, as well as ovarian carcinomas [176]. The secondary Müllerian system, which refers to the presence of Müllerian epithelium outside the indigenous locations, has been proposed [177] and this system may be an origin of ovarian carcinomas [178], endometriosis [179], endosalpingiosis [180], and retroperitoneal carcinomas [9,31,45,54]. However, this notion remains unverified [176]. Coelomic metaplasia may have been considered the most plausible hypothesis of retroperitoneal carcinogenesis, but direct evidence may not exist to support this theory at present. However, as ovarian serous carcinomas originate from both fallopian tube epithelium and ovarian surface epithelium [84,88,89], there is a possibility that retroperitoneal carcinomas develop directly from the abdominal and pelvic peritoneum that invaginates into the retroperitoneal space.

Endosalpingiosis and endometriosis are both involved in the development of ovarian, peritoneal, and retroperitoneal carcinomas [81,95,124,125,181,182]. Nevertheless, the pathogenesis of these diseases, in particular endosalpingiosis, is poorly understood. Although endosalpingiosis resembles tubal epithelium, it is associated with the development of not only serous tumors, but also mucinous carcinomas [81]. Endosalpingiosis may be a hereditary disease, similar to endometriosis [183], as endosalpingiosis is associated with *BRCA* mutation [124]. In addition, patients with endosalpingiosis had lower overall survival than patients with endometriosis [81]. Are shed epithelial cells of the fallopian tube a cause of endosalpingiosis and/or a precursor of both peritoneal and retroperitoneal carcinomas? If so, what is the difference between cells that become a peritoneal carcinoma and those that become a retroperitoneal carcinoma? Further studies on the pathogenesis of retroperitoneal, peritoneal, and ovarian carcinomas are necessary, as well as on endosalpingiosis and endometriosis.

## Figures and Tables

**Figure 1 cancers-15-04614-f001:**
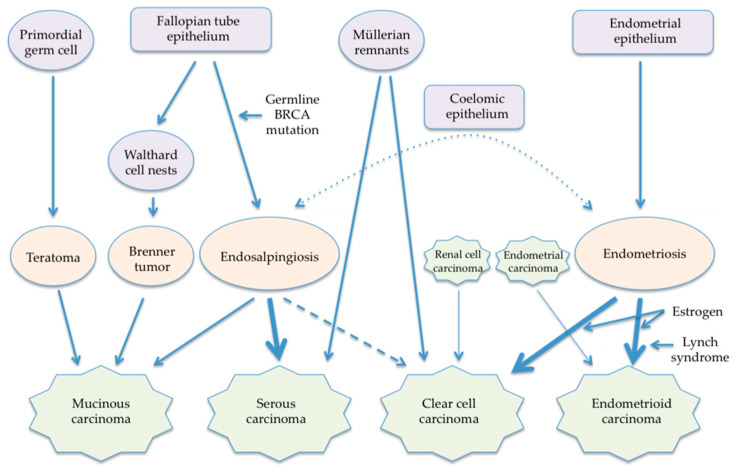
Newly proposed pathogenesis of retroperitoneal carcinomas.

**Figure 2 cancers-15-04614-f002:**
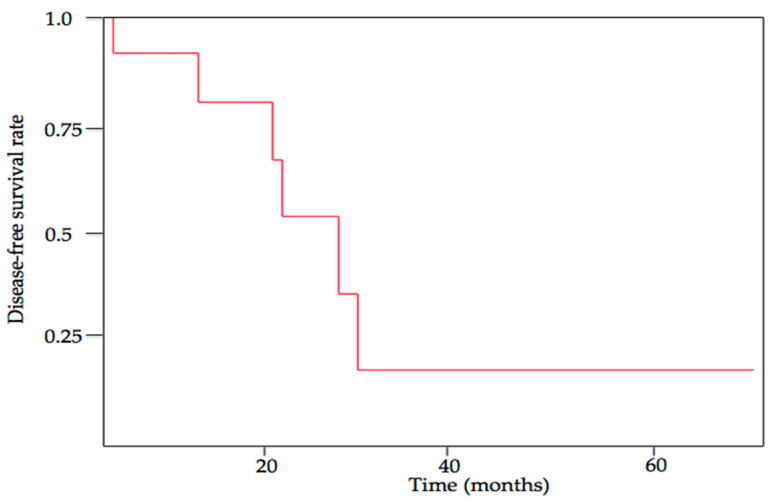
Kaplan–Meier disease-free survival curve for retroperitoneal serous carcinoma.

**Table 1 cancers-15-04614-t001:** Characteristics of retroperitoneal mucinous carcinoma.

Characteristics	Description
Sex	Female, 94% (73/78): Male, 6% (5/78)
Median age	Female, 42.0 years; Male, 62.2 years
Symptoms	A palpable abdominal mass, 42.9%
	Abdominal pain, 23.8%
Management	Surgery (100%)
	Adjuvant chemotherapy 24.1% (13/72)
Outcomes	Recurrence 40.4% (median, 8 months; 23/57)
	Median disease-free survival, 15 months (range, 1–130 months)
	5-year disease-specific survival, 75.4%
	Data from Myriokefalitaki et al., 2016 [8]

**Table 2 cancers-15-04614-t002:** (Primary) retroperitoneal serous carcinomas reported in the literature.

	Author	Age	Sex	Location	CA125 (U/mL)	Other Diseases and Health Conditions	Histology	Treatment	Outcomes
1	Ulbright et al., 1983 [14]	11	F	Encased the right common iliac vessels	NA	NA	Serous	Surgery (incomplete), Chemo (CAP), RT	NED, 10 months
2	Caruncho, et al., 1993 [15]	49	F	Surrounding the anterior aspects of the aorta	NA	Previous H/BSO for uterine leiomyoma	Serous	Surgery (incomplete)	NA
3	Kurosaki et al., 1998 [16]	38	F	Adherent to the lower pole of the right kidney	NA	NA	Serous	Surgery (complete)	NED, 24 months
4	Fujiwara et al., 1999 [17]	54	F	Para-aortic region	4000	None	Serous	Chemo (CP), Surgery# (incomplete), RT	DOD, 24 months
5	Kaku et al., 2004 [18]	44	F	Para-aortic region	218	Breast ca (past), LN met, Peritoneal recurrence of serous ca	Serous, G1	Surgery (complete)	AWD, 23 months
6	Demir et al., 2007 [19]	40	F	Right suprarenal fossa	NA	Diaphragmatic implants	Serous	Surgery (complete)	NA
7	Iura et al., 2009 [20]	66	F	Adherent to the ileocecum (para-aortic to iliac region)	678	Adnexal serous ca (concurrent), peritoneal carcinomatosis	HGSC	Surgery# (incomplete), Chemo (TC)	AWD, 32 months
8	Arichi et al., 2011 [21]	75	F	Between the right kidney and liver	335	None	HGSC	Surgery (complete), Chemo (DC)	NED, 6 months
9	Zhang et al., 2017 [22]	58	F	Right side of the Douglas pouch	3942	None	HGSC	Surgery# (complete), Chemo (TC)	AWD, 30 months
10	Kohada et al., 2017 [23]	42	F	Adjacent to the lower pole of the left kidney	10	None	HGSC *	Surgery (complete), Chemo (TC)	NED, 5 months
11	Nakatake et al., 2018 [24]	74	F	Adjacent to the liver (S6)	NA	Liver met	HGSC *	Surgery (complete)	NED, 12 months
12	Chae et al., 2019 [25]	71	M	Invading right psoas muscle and 12th rib	NA	Prostate ca (past)	HGSC *	Chemo (TC), RT, immunotherapy (Niv)	AWD, 15 months
13	Suda et al., 2019 [26]	58	F	Mesorectum	315.2	Concurrent STIC	HGSC *	Surgery# (complete), Chemo (TC + Bev)	NED, 20 months
14	Zhou et al., 2021 [28]	62	F	Douglas cul-de-sac	75	NA	HGSC *	Surgery#, Chemo (TC)	NED, NA
15	Win et al., 2021 [27]	60	F	Left pelvic cavity	253	Endometriosis	HGSC *	Surgery (incomplete), Chemo (TP)	DOD, 6 months
16	Otsuka et al., 2023 [29]	62	F	Common iliac to lower para-aortic region	2789	Uterine leiomyoma, LN met	HGSC *	Surgery# (optimal), Chemo (TC)	NED, 74 months

F, female. M, male. NA, not available. NED, no evidence of disease. DOD, dead of disease. AWD, alive with disease. Surgery#, surgery including bilateral salpingo-oophorectomy, CAP, cyclophosphamide, Adriamycin, cisplatin. TC, paclitaxel, carboplatin. DC, docetaxel, carboplatin. RT, radiation therapy, Niv, nivolmab. Bev, bevacizumab. * p53 abnormality was observed. STIC, serous tubal intraepithelial carcinoma.

**Table 3 cancers-15-04614-t003:** (Primary) retroperitoneal endometrioid and clear cell carcinomas reported in the literature.

	Author	Age	Sex	Location	Other Diseases and Health Conditions	Histology	Treatment	Outcomes
Endometrioid carcinoma								
1	Salerno et al., 2005 [30]	54	F	Periureter	Endometriosis (periureteral), previous H/BSO (serous cyst), ERT	Endometrioid	Surgery, Chemo (Carbo, LD)	NED 12 months
2	Tanaka et al., 2014 [31]	52	F	Left side of the bladder	Malignant lymphoma (past), Endometriosis (previous H/BSO), ERT	Endometrioid	Surgery, Chemo	NED 3 months
3	Osaku et al., 2019 [32]	52	F	Periureter (adjacent the left ilial vessels)	Endometriosis	Endometrioid	Chemo (Carbo Pacli), Surgery#, Chemo (Carbo, LD)	NED 24 months
4	Koual et al., 2021 [33]	50	F	External ilial lymph node	Lynch syndrome, previous cesarean section	Endometrioid	Surgery#, Chemo (Carbo, Pacli), RT	NED 36 months
5	Fischerova et al., 2023 [34]	29	F	Iliac lymph nodes	Lynch syndrome, Multiple sclerosis	Endometrioid	Surgery#, Chemo (Carbo, Pacli)	NA
Clear cell carcinoma								
1	Brooks et al., 1977 [35]	48	F	Anterior to the right ureter	Endometriosis (previous H/BSO), ERT	Clear cell	Surgery, RT	NED 22 months
2	Shields et al., 2020 [36]	71	M	Left para-aortic lymph node	Benign prostatic hypertrophy	Clear cell	Surgery	NED 60 months
3	Vatansever et al., 2021 [37]	37	F	Around the left internal iliac vein	None	Clear cell	Surgery# (complete), Chemo (Carbo, Pacli), RT	NED 41 months
4	Zeng et al., 2022 [38]	46	M	Near the left renal hilum	None	Clear cell	Surgery	Recurrence 6 months
Mixed (Clear cell carcinoma, endometrioid, mucinous) carcinoma								
1	Elnemr et al., 2010 [44]	46	F	Left retroperitoneal cavity	None	Serous, mucinous, endometrioid	Surgery, Chemo (IP Cis, Doce)	Recurrence 4 months
2	Vatansever et al., 2021 [37]	45	F	Around the left external iliac vein	Previous H/BSO (for unknown reason)	Clear cell, endometrioid	Surgery (partial), Chemo (Carbo, Pacli), RT	NED 30 months

F, female. M, male. NA, not available. H/BSO, hysterectomy and bilateral salpingo-oophorectomy. Surgery#, surgery including bilateral salpingo-oophorectomy. ERT, estrogen replacement therapy. Carbo, carboplatin. Pacli, paclitaxel. LD, liposomal doxorubicin. RT, radiation therapy. IP, intraperitoneal. Cis, cisplatin. Doce, docetaxel. NED, no evidence of disease.

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
