# Peer review of "Primary Retroperitoneal Carcinomas: New Insights into Pathogenesis and Clinical Management in Comparison with Ovarian Carcinomas and Carcinoma of Unknown Primary"

_cancers, 2023, doi:10.3390/cancers15184614_

Round 1

Reviewer 1 Report

This paper is well written and thoroughly covers the area. I would have liked a more thorough comparison between types showing similarities and differences and an explaination of why the differences make it difficult to treat. This will be instead of each individual type being explained.

Author Response

Thank you for your important comments. I have thought the same thing about ovarian cancers. Can we classify ovarian cancers into several categories with an integrated genomic, transcriptomic and proteomic characterization, like endometrial cancer in a TCGA project [Nature 2013]? However, ovarian cancers analyzed in a TCGA project included only high-grade serous carcinoma [Nature 2011], so we do not have sufficient information on non-serous ovarian carcinomas, such as endometrioid, clear cell, and mucinous carcinomas. Unfortunately, as primary retroperitoneal carcinomas are rare occurrences, information on molecular aberrations that cause these carcinomas is lacking. An accumulation of cases may answer our question.

Reviewer 2 Report

The review manuscript “Primary retroperitoneal carcinomas: new insights into pathogenesis and clinical management in comparison with ovarian carcinomas and carcinoma of unknown primary” by Isao Otsuka to discuss the hypotheses about pathogenesis of retroperitoneal carcinomas. Minor concern that mustbe taken into account before the work can be reconsidered for publication.

Comment

Figure 1: Other gene mutations such as: p53, PI3K should be added. Not only BRCA1.

Author Response

Thank you for your pertinent comment. In Figure 1, “BRCA mutation” was insufficient. “Germline BRCA mutation” is correct (Please see 5.2 and references 124), and I have changed the words. p53 and PI3K have not been reported to be associated with the development of endosalpingiosis.

Reviewer 3 Report

Fairly interesting review about retroperitoneal carcinomas.

Author Response

Thank you for your comments.

Reviewer 4 Report

In this article, the authors discussed the pathogenesis of primary retroperitoneal carcinomas based on the knowledge of ovarian carcinogenesis. They also explored the clinical management of retroperitoneal carcinomas. The manuscript is straightforward, well written, and concise and has clear results within the scope of a review article. Definitely deserves to be published and is a valuable contribution to the “cancers” journal. However, the following comments need to be addressed, as recommended.

[1] “1. Introduction”, Page 2 of 19, Lines 46-50:

Pathogenesis of primary retroperitoneal carcinomas remains unknown. Their pathogenesis, however, may be associated with that of ovarian carcinomas, as histological subtypes of retroperitoneal carcinomas are the same as those of ovarian carcinomas. In addition, primary retroperitoneal carcinomas almost exclusively develop in women, excluding mucinous tumors that develop both in women and men.”

Furthermore, it should be reported that – based on cross-sectional imagingthe differential diagnosis for primary retroperitoneal carcinomas may include malignant peritoneal mesothelioma, peritoneal carcinomatosis, serous peritoneal, and ovarian carcinoma, as well as lymphomatosis and tuberculous peritonitis. Biopsy is essential for establishing diagnosis and can be performed either radiographically or surgically. Laparoscopy represents a preferable diagnostic approach, considering its lower invasiveness and clear intraoperative assessment. Whereas the wall of the mobile small bowel is typically not involved in peritoneal carcinomatosis, during either laparoscopy or laparotomy in patients with no medical history of abdominal operations, the serosal layer of the small bowel wall, consisting of mesothelial cells, is commonly diffusely involved in malignant peritoneal mesothelioma. The mesentery is involved in both cases.

[2] “5.5. Carcinosarcoma”, Page 10 of 19, Lines 355-356:

Retroperitoneal carcinosarcoma, as well as uterine carcinosarcoma [108], may derive from serous or endometrioid carcinoma.”.

At that stage, the authors should also mention that – similarly to retroperitoneal and uterine carcinosarcoma – the more frequently encountered epithelial components in ovarian carcinosarcomas are serous, endometrioid, or undifferentiated adenocarcinoma. Specifically, ovarian carcinosarcomas are classified according to the homologous or heterologous derivation of the mesenchymal tissue in their stromal element. Homologous carcinosarcomas contain sarcomatous elements differentiate towards tissues physiologically native to the ovary and include fibrosarcoma, and leiomyosarcoma. The heterologous components usually contain malignant osteoid, chondroid or rhabdomyoid cells that are physiologically foreign to the primary site.

Recommended reference: Boussios S, et al. Ovarian carcinosarcoma: Current developments and future perspectives. Crit Rev Oncol Hematol. 2019;134:46-55.

[3] “6. Retroperitoneal carcinoma as a part of carcinoma of unknown primary”, Page 10 of 19, Lines 366-368:

Carcinoma of unknown primary represents a heterogeneous group of metastatic tumors for which a standard diagnostic work-up fails to identify the site of origin at the time of diagnosis [156]”.

Please, do report that patients with carcinoma of unknown primary (CUP) are categorized into two prognostic subgroups, according to their clinicopathologic characteristics. The majority of patients with CUP (80–85%) belong to unfavorable subsets. In this subset, two prognostic groups are identified according to the performance status and LDH level. The favorable risk cancer subgroup (15–20%) includes patients with neuroendocrine carcinomas of unknown primary, peritoneal adenocarcinomatosis of a serous papillary subtype, isolated axillary nodal metastases in females, squamous cell carcinoma involving non-supraclavicular cervical lymph nodes, single metastatic deposit from unknown primary and men with blastic bone metastases and PSA expression. Very recently, new favorable subsets of CUP seem to emerge including colorectal, lung and renal CUP which underlies specific treatments.

Recommended reference: Rassy E, et al. New rising entities in cancer of unknown primary: Is there a real therapeutic benefit? Crit Rev Oncol Hematol. 2020 Mar;147:102882.

[4] “7.2. Treatment”, Page 11 of 19, Lines 419-422:

In non-mucinous carcinomas, a retroperitoneal tumor may not be primary but metastasis from clinically undetectable primary tumors, thus adjuvant treatments should be considered.”.

In that point, the authors are highly encouraged to report that there are significant deficiencies in the currently available studies comparing site-specific therapy and empiric chemotherapy for cancers of unknown primary site (CUP). These deficiencies include patient accrual problems (oversampling treatment-resistant tumor types and long recruitment), study design limitations (observational and problematic trials), heterogeneity among the CUP classifiers (epigenetic vs. Transcriptomic profiling), and incomparable therapies. The assessment of recently published CUP literature allows to recommend two comprehensive clinical trial designs, a visionary and a pragmatic approach. Both are amenable to implementing the latest diagnostics and therapeutic advances to improve the quality of CUP research and the prognosis of many patients.

[5] “7.3. Prognosis”, Page 11 of 19, Lines 431-434:

Mucinous tumors are usually associated with a favorable prognosis, except those with mural nodules that are often diagnosed as sarcoma, anaplastic carcinoma, or carcinosarcoma [170,171].”.

At that stage, the authors should specify the case of the sarcomas of undefinied primary (SUP). The pathologies of the patients diagnosed with SUP are highly variable reflecting a histologic heterogeneity that complicates diagnostic confirmations. The possible theoretical explanation model for the SUP phenomenon is the smallness of the primary tumor that evades detection or inadequate workup. It has been reported that two-thirds of the patients have more than one metastatic site.

English language is in a good scientifically level.

Author Response

Thank you for your valuable comments. This narrative review explores pathogenesis of primary retroperitoneal carcinomas (PRC) based on the knowledge of ovarian carcinogenesis and discusses clinical management of retroperitoneal carcinomas. From these perspectives, I respond to the reviewer’s comments.

[1] These comments are appropriate for carcinoma of unknown primary (CUP), but not for PRC. PRC is rarely associated with intraperitoneal fluid and/or peritonitis, including peritonitis carcinomatosa. Tumor biopsy that may cause dissemination of tumor cells into the peritoneal cavity is not usually performed in PRC, similar to early-stage ovarian carcinoma. I agree that laparoscopy is a useful diagnostic approach. However, different from CUP, total removal of the tumor is the mainstay of treatment for PRC. The effectiveness and safety of laparoscopic resection of PRC has not been established.

[2] I have added a comment on ovarian carcinosarcoma in the manuscript. However, as only three cases of retroperitoneal carcinosarcoma have been reported, the description of ovarian carcinosarcomas may not apply to retroperitoneal carcinosarcoma. Accumulation of cases is required to make relevant comments.

[3] I have added some of your comments on CUP in the manuscript. However, cancers less relevant to PRC, such as peritoneal carcinomatosis, and squamous cell carcinoma that is not discussed in this manuscript, are not added.

[4] As this manuscript is on PRC and not on CUP, too detailed descriptions of the treatment for CUP are not appropriate. Part of CUP may be a PRC as written in the manuscript, and “site-specific” therapy for PRC has not been established.

[5] Comments on the sarcomas of unknown primary (SUP) are beyond the scope of this paper that discusses PR carcinoma.

Round 2

Reviewer 1 Report

The only major concern I have with the article is Figure 1. Why did the colors change from the original version? I find it harder to read now.